# High-Risk Early-Stage Endometrial Cancer: Role of Adjuvant Therapy and Prognostic Factors Affecting Survival

**DOI:** 10.3390/cancers17122056

**Published:** 2025-06-19

**Authors:** Ji Hyun Hong, Jun Kang, Sung Jong Lee, Keun Ho Lee, Soo Young Hur, Yeon-Sil Kim

**Affiliations:** 1Department of Radiation Oncology, Seoul St. Mary’s Hospital, College of Medicine, The Catholic University of Korea, Seoul 06591, Republic of Korea; jihyunhong@catholic.ac.kr; 2Department of Hospital Pathology, Seoul St. Mary’s Hospital, College of Medicine, The Catholic University of Korea, Seoul 06591, Republic of Korea; 3Department of Obstetrics and Gynecology, Seoul St. Mary’s Hospital, College of Medicine, The Catholic University of Korea, Seoul 06591, Republic of Korea

**Keywords:** high-grade endometrial cancer, adjuvant therapy, radiotherapy, chemotherapy, prognostic factors

## Abstract

High-grade endometrial cancer, including non-endometrioid and grade 3 endometrioid types, shows a poor prognosis with unclear adjuvant treatment strategies in early-stage disease. Although the revised staging system integrates both pathological and molecular features, complete molecular profiling remains challenging to implement routinely in real-world clinical practice. Non-endometrioid histology correlates with distant metastasis, whereas grade 3 endometrioid carcinoma shows a higher risk of locoregional failure; deep invasion, lack of lymphadenectomy, and omission of adjuvant therapy were identified as adverse prognostic factors. These results highlight the importance of the selective use of adjuvant therapy in early-stage high-grade cases with high-risk features.

## 1. Introduction

Endometrial cancer is one of the most prevalent gynecological malignancies. The overall prognosis is favorable for patients with early-stage endometrial cancer with low-grade disease [1]. However, several factors, including the completeness of surgical resection, histological, molecular subtype, lymphovascular space invasion, and myometrial invasion, can affect the risk of recurrence and prognosis.

Endometrioid and non-endometrioid endometrial cancer are two pathological entities distinguished by histological differences. Non-endometrioid endometrial cancer is a rare neoplasm that accounts for approximately 20% of all cases with aggressive features and poor prognosis [2]. Moreover, the incidence rate has rapidly increased [3]. Non-endometrioid endometrial cancer is frequently compared to grade 3 endometrioid carcinoma and is considered to have similar outcomes. To elucidate the pathological and molecular features, the International Federation of Gynecology and Obstetrics (FIGO) revised the staging system in 2023 [4], identifying grade 3 endometrioid carcinoma and non-endometrioid endometrial cancer as aggressive histological subtypes.

The mainstay of management is surgery, including total hysterectomy and bilateral salpingo-oophorectomy with or without lymph node dissection. Adjuvant therapy is decided based on the pathological evaluation, and radiotherapy (RT) with or without chemotherapy has been recommended for patients with a high risk of recurrence [2,5,6]. However, controversies remain regarding the optimal adjuvant treatment for early-stage high-grade disease, owing to its rarity and heterogeneity. With the increasing rates of non-endometrioid endometrial cancer and revised FIGO staging, the importance of identifying the optimal adjuvant therapy for early-stage high-grade disease has surged. Therefore, in this study, we investigated the survival outcomes of patients with early-stage high-grade disease and aimed to identify the prognostic factors affecting survival. Additionally, we aimed to identify the optimal patient candidates for adjuvant therapy.

## 2. Materials and Methods

The medical records of 345 female patients who underwent hysterectomy for FIGO stage I and II endometrial cancer at a single institution between September 2008 and December 2022 were reviewed. Patients who were pathologically confirmed to have high-risk non-endometrioid and grade 3 endometrioid carcinoma, were included in the eligibility criteria. Patients diagnosed with grades 1 and 2 endometrioid carcinoma (n = 205), coexisting cancers other than breast and thyroid cancers (n = 11), insufficient records (n = 19), and mucinous histology (n = 4) were excluded from the analysis. Finally, 106 patients were included in this study. All the procedures performed in studies involving human participants were in accordance with the ethical standards of the institutional and/or national research committee and with the Helsinki Declaration of 1975, as revised in 1983. This study was reviewed and approved by the Institutional Review Board of Seoul St. Mary’s Hospital (Number: KC24RISI0174) on 12 March 2024. Because the study was retrospective, the requirement for patient consent was waived.

All the patients underwent hysterectomy and salpingo-oophorectomy. Lymph nodal staging, peritoneal washing, and omental evaluations were performed at the discretion of the surgeon. A multidisciplinary team determined the adjuvant treatment based on the clinical and pathologic findings. The most common chemotherapy regimen was carboplatin/paclitaxel. All external beam radiation therapy (EBRT) was administered to the entire pelvis, with a dose range of 45–56 Gy delivered in 1.8–2 Gy daily fractions. Intracavitary radiation (ICR) was delivered using high-dose-rate brachytherapy to the stump.

Descriptive statistics were used to analyze the patient characteristics and treatments. These characteristics were compared between non-endometrioid and grade endometrioid carcinoma using Fisher’s exact test and the Mann–Whitney U test. Survival analyses were performed using the Kaplan–Meier method, and the resulting data were evaluated using the log-rank test. The date of origin of all the survival outcomes was defined as the hysterectomy date. The follow-up duration for each patient was defined as the time from the date of hysterectomy to the date of death, censoring, or last follow-up. The median follow-up was calculated using the reverse Kaplan–Meier method. Overall survival (OS) and disease-free survival (DFS) were evaluated from the date of hysterectomy to the date of death or disease recurrence, respectively, or to the date of the last follow-up, if no event occurred. Locoregional recurrence-free survival (LRRFS) was defined as the time from hysterectomy to pelvic or vaginal recurrence. Distant metastasis-free survival (DMFS) was defined as the time from hysterectomy to the occurrence of evidence of distant metastasis.

The prognostic factors associated with survival outcomes were analyzed using Cox proportional hazards regression models. Multivariate analysis was performed for survival outcomes using backward selection, with covariates that had a *p*-value of less than 0.1 in the univariate analysis. Statistical significance was set at *p* < 0.05. All the statistical analyses were performed using the STATA/SE software (version 17.0; StataCorp, LLC, College Station, TX, USA) and the R program version 4.5.0 (R Development Core Team, Vienna, Austria).

## 3. Results

Among all the included patients, 60 (56.7%) were pathologically confirmed to have non-endometrioid endometrial cancer, while 46 (43.3%) were diagnosed with grade 3 endometrioid carcinoma. The characteristics of the patients and tumors are presented in Table 1. Among the patients with non-endometrioid endometrial cancer, the most common histological type was serous carcinoma, followed by carcinosarcoma, mixed carcinoma, clear cell carcinoma, and undifferentiated carcinoma. The median age at hysterectomy was 61 years (range, 33 to 90 years). The median age of patients with non-endometrioid endometrial cancer was significantly higher (65 vs. 57 years, *p* = 0.001) than that of patients with grade 3 endometrioid carcinoma. Among all patients, 66 (62.3%) were diagnosed with FIGO stage IA, 26 (24.5%) with FIGO stage IB, and 15 (14.2%) with FIGO stage II disease. A total of 73 patients (68.9%) underwent minimally invasive surgery, including laparoscopic or robotic hysterectomy. During the surgical procedure, 62 (58.5%) and 17 (16.0%) patients underwent proper peritoneal washing and omental evaluation, respectively. Sentinel lymph node biopsy was performed in 17 (28.3%) with non-endometrioid endometrial cancer and in 11 (23.9%) patients with grade 3 endometrioid carcinoma. Appropriate pelvic lymph node dissection (PLND), defined as the removal of >10 lymph nodes based on their early stage, was observed at a greater frequency in patients with grade 3 endometrioid carcinoma (60.0% vs. 82.6%, *p* = 0.04) than in those with non-endometrioid endometrial cancer.

A total of 69 patients (65.1%) received adjuvant treatment. The most prevalent modality was RT alone (38 patients, 35.9%), followed by chemotherapy and RT (18 patients, 17.0%). Adjuvant RT was administered to 56 patients (52.8%). Among these patients, forty-five received EBRT alone, three received both EBRT and ICR, and eight received ICR alone. Adjuvant chemotherapy was administered to 33 patients (31.1%), with a median of six cycles. Patients with non-endometrioid endometrial cancer received adjuvant chemotherapy more frequently than those with grade 3 endometrioid carcinoma (51.7% vs. 4.3%, *p* < 0.001). A total of 36 patients (34.0%) did not undergo any adjuvant treatment, in addition to active surveillance.

After a median follow-up period of 48.8 months (range of 2.1 to 185.5 months), disease progressed in 37 patients (34.9%), and 21 patients (19.8%) died. The median time to the first progression was 12.9 months (range of 2.6 to 68.0 months). Thirty (28.3%) and 22 (20.8%) patients experienced distant metastases and locoregional recurrence, respectively (Appendix A). The most prevalent distant metastatic site was peritoneal seeding, followed by the lungs (Appendix A). Patients with non-endometrioid and grade 3 endometrioid carcinoma showed disease progression in 35.0% (n = 21) and 34.8% (n = 16) of cases, respectively (Appendix A). With similar disease progression rates, a higher proportion of distant metastases was observed in patients with non-endometrioid than in those with grade 3 endometrioid carcinoma (31.7% vs. 23.9%, *p* = 0.38), whereas a higher proportion of locoregional failure was observed in patients with grade 3 than in those with non-endometrioid endometrial cancer (18.3% vs. 23.9%, *p* = 0.48).

The 3-year OS, DFS, LRRFS, and DMFS rates were 86.0%, 65.8%, 79.6%, and 72.1%, respectively. Compared with grade 3 endometrioid carcinoma, non-endometrioid endometrial cancer showed significantly inferior OS (Figure 1A, *p* = 0.002). The survival outcomes differed between the two groups of patients who received adjuvant therapy and those who did not (Figure 2). The patients who received adjuvant therapy showed superior OS (*p* = 0.011) and LRRFS (*p* = 0.019) with statistical significance. The differences in survival outcomes and failure patterns according to adjuvant treatment type are summarized in Appendix A.

In the univariate analysis, non-endometrioid endometrial cancer (*p* = 0.005), less than 10 lymph nodes’ dissection (*p* < 0.001), higher depth of invasion (DOI) (*p* < 0.001), and absence of adjuvant treatment (*p* = 0.016) were unfavorable prognostic factors associated with OS (Table 2). Additionally, a higher body mass index (BMI) (*p* = 0.003), positive washing results (*p* = 0.017), and pathologic omental evaluation (*p* = 0.018) were identified as significant prognostic factors associated with LRRFS, DFS, and DMFS, respectively. In the multivariate analysis, FIGO stage II was identified as an unfavorable factor for OS (*p* = 0.043), while a higher BMI was found to be an unfavorable factor for LRRFS (*p* = 0.021) (Table 3).

The risk evaluation was divided into four groups based on the unfavorable prognostic factors for OS, as identified through univariate analysis, including the absence of PLND, non-endometrioid endometrial cancer, and a higher DOI. Group 1 included patients with all the favorable factors (grade 3 endometrioid carcinoma, DOI < 25%, with PLND). Group 2 had one unfavorable factor (one of non-endometrioid endometrial cancer, a DOI ≥ 25%, or without PLND). Group 3 had two unfavorable factors (two of non-endometrioid endometrial cancer, a DOI ≥ 25%, or without PLND). Group 4 included patients with all the unfavorable factors (non-endometrioid endometrial cancer, a DOI ≥ 25%, without PLND). Subgroup analysis was conducted according to the risk groups. The 3-year OS, DFS, LRRFS, and DMFS rates for patients without any unfavorable factors (group 1) were 100%, 69.2%, 83.3%, and 69.2%, respectively (Appendix A). A difference was observed in the OS between the risk groups (*p* = 0.135) without statistical significance. When the patients in groups 2 and 3 were combined, adjuvant treatment was associated with better survival outcomes, including OS (*p* = 0.009), DFS (*p* = 0.021), and LRRFS (*p* = 0.034) (Appendix A). In contrast, the other group exhibited no significant differences.

## 4. Discussion

We investigated the survival outcomes and prognostic factors in patients with FIGO stage I to II high-grade endometrial cancer, specifically focusing on non-endometrioid and grade 3 endometrioid carcinoma. The study demonstrated an overall 5-year OS rate of 78.2%. Patients with non-endometrioid endometrial cancer had a significantly shorter OS (64.7%) compared to those with grade 3 endometrioid carcinoma (92.4%) (*p* = 0.002). Adjuvant therapy was associated with a superior OS (*p* = 0.011) and LRRFS (*p* = 0.019). The FIGO stage, lymphadenectomy involving >10 lymph nodes, and the depth of invasion were significant predictive factors for the OS, underscoring the importance of surgical staging based on lymphadenectomy.

Although recent studies have aimed to identify the differences in survival outcomes and the efficacy of adjuvant treatments for high-grade early-stage disease [7], studies addressing them are still lacking. Clinical guidelines recommend brachytherapy alone for patients at low, intermediate, and high-intermediate risk at the ESMO-ESGO-ESTRO consensus conference [6,8,9]. However, more aggressive adjuvant treatments, such as EBRT with or without brachytherapy, are often administered to high-risk stage I patients in real-world practice. Our study found that patients with non-endometrioid endometrial cancer had an inferior OS and received more heterogeneous adjuvant therapies than patients with grade 3 endometrioid carcinoma. Despite a higher proportion of patients with non-endometrioid endometrial cancer receiving chemotherapy, they showed inferior DMFS compared with grade 3 endometrioid carcinoma, although the difference was not statistically significant.

### 4.1. Survival Outcomes and Histological Subtypes

Previous studies have reported a wide range of survival outcomes for early-stage non-endometrioid and grade 3 endometrioid carcinoma, with the 5-year OS ranging from 55% to 93.4% and from 66% to 94.6%, respectively [10]. This study demonstrated comparable survival outcomes, with an overall 5-year OS rate of 78.2%.

Several studies have reported that different subtypes have different survival outcomes [7]. For example, carcinosarcoma has a 5-year OS of 36.0 to 64.6% and a 5-year DFS of 46.4% to 57.5% [11,12,13], while serous carcinoma and clear cell carcinoma demonstrated a better OS and DFS (range of 71.0 to 79.0% and 66.0 to 75.0%, respectively) [11,14]. Furthermore, recent studies [15,16] on the molecular subgroups defined by the Cancer Genome Atlas project [17] have shown clear differences in prognosis between molecular groups. The wide range of survival outcomes may be attributed to the histological and molecular heterogeneity of high-grade endometrial cancer. Although the importance of molecular profiling has been increasingly recognized and the research in this area is expanding [18,19,20], its routine implementation in real-world clinical practice remains limited. Our study uniquely included a broader spectrum of non-endometrioid subtypes, thus enhancing the understanding of their individual and collective impact on the treatment outcomes.

We observed that patients with non-endometrioid endometrial cancer had a significantly shorter OS than those with grade 3 endometrioid carcinoma (*p* = 0.002). Several studies have shown consistent survival results [10,21], whereas a few have reported no significant differences [22]. Moreover, higher rates of distant metastasis were observed in non-endometrioid endometrial cancer than in grade 3 endometrioid carcinoma, whereas locoregional failure was more common in grade 3 endometrioid carcinoma. Previous studies have also highlighted a higher risk of distant metastasis in non-endometrioid endometrial cancer, suggesting the necessity for adjuvant chemotherapy [21].

### 4.2. Role of Adjuvant Therapy and Prognostic Factors

Although the optimal treatment remains controversial, several studies have underscored the importance of adjuvant treatment in early-stage high-grade disease [23]. For example, adjuvant chemotherapy and brachytherapy have shown beneficial effects in patients with early-stage serous carcinoma [24,25]. The results for patients with early-stage clear cell carcinoma are more variable [23,24,25,26,27]; while Nieto et al. [23] and Hong et al. [24] did not show any meaningful survival benefit from adjuvant therapy, Chang-Halpenny et al. [26], Xiang et al. [25], and Kim et al. [27] reported positive effects of chemotherapy and RT including brachytherapy. Although our study did not show statistically significant differences specifically for EBRT or chemotherapy, the presence of adjuvant therapy improved the OS and LRRFS.

In this study, FIGO stage II was an unfavorable prognostic factor for the OS, consistent with a population-based cohort study by Akesson et al. [28]. Significant controversy persists regarding the extent of lymph node dissection and the role of sentinel lymph node (SLN) biopsy in early-stage endometrial cancer. In our study, lymphadenectomy involving >10 lymph nodes and the depth of invasion were significant predictive factors for the OS in the univariate analysis. Some studies support SLN mapping followed by full lymphadenectomy [29,30,31,32], and the NCCN guidelines recommend SLN biopsy for the surgical staging of endometrial cancer regardless of the risk group. However, concerns regarding its safety remain because of its therapeutic role. The retrospective SEPAL study showed a survival benefit for systemic pelvic and para-aortic lymphadenectomy in the high-risk group [33], and the ESMO-ESGO-ESTRO guidelines support lymphadenectomy as part of comprehensive staging [6]. Consistent with other studies [34,35], the depth of invasion was identified as a predictive factor in our study. Frei et al. [36] and Lee et al. [37] also highlighted the depth of invasion as a predictive factor for nodal metastasis, indicating the need for extensive surgical lymph node assessment in patients with deeper myometrial invasion.

Based on the three factors identified in the univariate analysis—histological type, PLND, and DOI—we conducted a subgroup analysis. Scores were assigned based on the presence of each adverse factor, stratifying patients into four risk groups. Patients with one or two unfavorable factors showed significant differences in the OS, DFS, and DMFS after adjuvant treatments. Therefore, patients with one or two unfavorable factors should consider adjuvant treatment to improve the survival outcomes. The higher risk of distant metastasis in non-endometrioid endometrial cancer and the higher prevalence of lymph node metastasis in patients with a greater depth of invasion, requiring more extensive surgical lymph node assessment, underscores the importance of adjuvant therapy. However, there remain controversial issues regarding stage IA non-endometrioid endometrial cancer confined to the endometrium without myometrial invasion [38,39].

This study is one of the few to focus on early-stage high-grade endometrial cancer. Given the necessity to develop effective treatment strategies for early-stage high-grade endometrial cancer, the findings of this study are important. They could contribute to establishing prognostic and therapeutic guidelines areas where the current comprehensive information is scarce.

This study has several limitations. First, the retrospective nature of the analysis included a heterogeneous cohort of patients, which may affect the survival outcomes due to the differences in clinical, histological, genetic, and molecular characteristics. Second, the surgical procedures were not standardized. Because of the retrospective nature of the study, patients with preoperative diagnoses suggesting lower-stage with FIGO grade 1–2 or adenocarcinoma in situ, or with imaging findings indicating disease confined to the uterus without lymph node enlargement, often did not undergo peritoneal washing and omental evaluation, or underwent only limited lymph node dissection. Nevertheless, some of these patients were ultimately diagnosed with high-grade early-stage endometrial cancer were included in the analysis. Third, heterogeneous treatment modalities can also affect the outcomes, as most treatments are decided by a multidisciplinary team based on individual patient characteristics, preventing the randomization of treatment modalities. Fourth, the lack of comprehensive molecular profiling in the real practice clinic limited its analysis based on molecular subtypes. In particular, POLE mutation testing was performed in only a small subset of patients, which restricted the evaluation of its prognostic significance. Finally, the statistical power was limited due to the small sample size.

This study can contribute to the clinical decision making regarding adjuvant treatment for early-stage high-grade endometrial cancer. Pathologic outcomes and surgical techniques are necessary to predict the prognosis and determine who need adjuvant treatment. With the revised FIGO staging, future research should investigate the role of molecular analysis in these patients. Given the rarity and heterogeneity of pathological entities, multicenter studies should be considered.

Authors should discuss the results and how they can be interpreted from the perspective of previous studies and of the working hypotheses. The findings and their implications should be discussed in the broadest context possible. Future research directions may also be highlighted.

## 5. Conclusions

In conclusion, this study indicates that patients with early-stage high-grade endometrial cancer, particularly those who present with one or more of the identified risk factors, including the non-endometrioid histologic subtype, absence of PLND, and a greater DOI, should be considered a high-risk group for progression and shortened survival, and adjuvant therapy should be recommended.

## Figures and Tables

**Figure 1 cancers-17-02056-f001:**
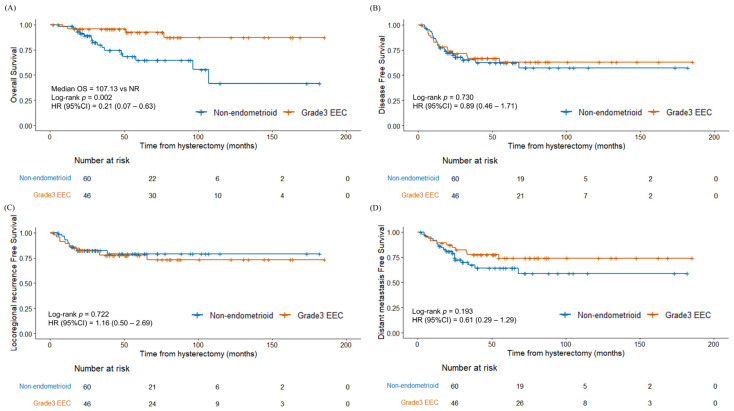
Kaplan–Meier survival curves between non-endometrioid endometrial cancer and grade 3 endometrioid carcinoma (**A**) overall survival, (**B**) disease-free survival, (**C**) locoregional recurrence-free survival, and (**D**) distant metastasis-free survival. Abbreviation: G3 EEC, grade 3 endometrioid carcinoma.

**Figure 2 cancers-17-02056-f002:**
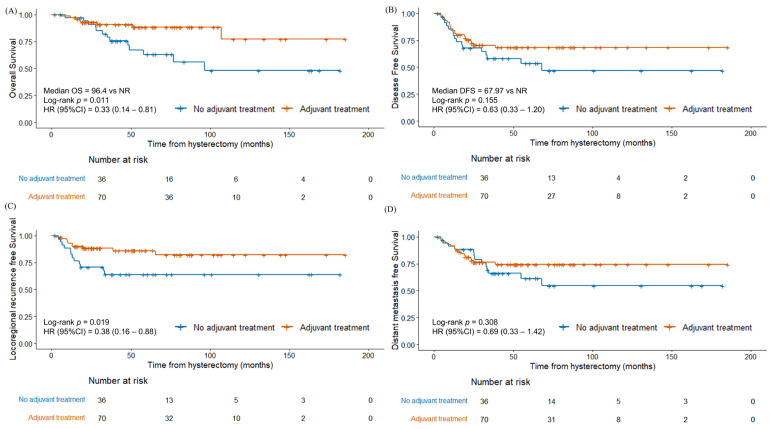
Kaplan–Meier survival curves between patients with adjuvant therapy and without adjuvant therapy (**A**) overall survival, (**B**) disease-free survival, (**C**) locoregional recurrence-free survival, and (**D**) distant metastasis-free survival.

**Table 1 cancers-17-02056-t001:** Patient and tumor characteristics.

Characteristic			Total (n = 106)	NEEC (n = 60)	G3 EEC (n = 46)	*p*-Value
BMI (median, range)			23.3 (17.0–37.7)	23.7 (19.2–31.5)	22.4 (17–37.7)	0.126
Age						
	<60 years		46 (43.4%)	17 (28.3%)	29 (63.0%)	<0.001
	≥60 years		60 (56.6%)	43 (71.7%)	17 (37.0%)	
Hypertension			31 (29.2%)	22 (36.7%)	9 (19.6%)	0.084
Diabetes			10 (9.4%)	7 (11.7%)	3 (6.5%)	0.508
History of breast cancer			13 (12.3%)	10 (16.7%)	3 (6.5%)	0.143
Histologic type						
		Serous carcinoma	23 (21.7%)	23 (38.4%)		
		Clear cell carcinoma	9 (8.5%)	9 (15%)		
		Carcinosarcoma	16 (15.1%)	16 (26.7%)		
		Undifferentiated	3 (2.8%)	3 (5%)		
		Mixed	9 (8.5%)	9 (15%)		
		Endometrioid carcinoma	46 (43.4%)		46 (100%)	
FIGO stage (2018)						0.004
	IA		66 (62.3%)	43 (71.7%)	23 (50.0%)	
	IB		26 (24.5%)	7 (11.7%)	19 (39.1%)	
	II		15 (14.2%)	10 (17.7%)	5 (10.9%)	
Minimally invasive surgery			73 (68.9%)	42 (70.0%)	31 (67.4%)	0.834
Peritoneal washing			62 (58.5%)	38 (64.4%)	24 (55.8%)	0.416
Omental evaluation						0.110
	No		85 (80.2%)	46 (78.0%)	39 (90.7%)	
	Omentectomy or Bx		17 (16.7%)	13 (22.0%)	4 (9.3%)	
SLNB			28 (26.4%)	17 (28.3%)	11 (23.9%)	0.662
PLND			88 (83.0%)	45 (75.0%)	43 (93.5%)	0.021
PaLND			44 (41.5%)	23 (38.3%)	21 (45.7%)	0.688
Dissected LN ≥ 10			74 (69.8%)	36 (60.0%)	38 (82.6%)	0.040
Tumor size (cm, median, range)			3.5 (0.0–8.8)	3.5 (0–8.8)	3.5 (0–8.4)	0.673
Myometrial invasion						0.100
	None		14 (13.2%)	9 (15.0%)	5 (10.9%)	
	<50%		52 (49.1%)	32 (53.3%)	20 (43.5%)	
	≥50%		31 (29.2%)	13 (21.7%)	18 (39.1%)	
DOI/thick (%, median, range)			22.8 (0.0–93.8)	21.4 (0–92.3)	28.6 (0–93.8)	0.266
LVSI			30 (28.3%)	13 (21.7%)	17 (37.0%)	0.193
Adjuvant treatment						<0.001
	None		36 (34.0%)	20 (33.3%)	16 (34.8%)	
	CTx alone		14 (13.2%)	12 (20.0%)	2 (4.3%)	
	RTx alone		37 (34.9%)	9 (15.0%)	28 (60.9%)	
	CTx-RTx		19 (17.9%)	19 (31.7%)	0 (0.0%)	
CTx						<0.001
	No		73 (68.9%)	29 (48.3%)	44 (95.7%)	
	Yes		33 (31.1%)	31 (51.7%)	2 (4.3%)	
RTx						0.199
	No		50 (47.2%)	32 (53.3%)	18 (39.1%)	
	Yes		56 (52.8%)	28 (46.7%)	28 (60.9%)	
RTx modality						0.225
	EBRT		45 (43.4%)	24 (40.0%)	21 (45.7%)	
	ICR		8 (7.6%)	2 (3.3%)	6 (13.0%)	
	EBRT + ICR		3 (2.8%)	2 (3.3%)	1 (2.2%)	

Abbreviations: BMI, body mass index; FIGO, International Federation of Gynecology and Obstetrics; Bx, biopsy; SLNB, sentinel lymph node biopsy; PLND, pelvic lymph node dissection; PaLND, para-aortic lymph node dissection; LN, lymph node; DOI, depth of invasion; LVSI, lymphovascular invasion; CTx, chemotherapy; RTx, radiotherapy; EBRT, external beam radiotherapy; ICR, intracavitary radiotherapy; NEEC, non-endometroid endometrial cancer; G3 EEC, grade 3 endometrioid carcinoma.

**Table 2 cancers-17-02056-t002:** Univariate survival analysis.

		OS			DFS			LRRFS			DMFS		
		HR	95% CI	*p* Value	HR	95% CI	*p* Value	HR	95% CI	*p* Value	HR	95% CI	*p* Value
BMI		1.09 (0.98~1.22)		0.104	1.09 (0.99~1.20)		0.092	1.19 (1.06~1.34)		0.003	1.06 (0.96~1.17)		0.217
Age	<60 * vs. ≥60	2.07 (0.83~5.15)		0.119	1.35 (0.70~2.61)		0.369	2.39 (0.93~6.14)		0.070	1.03 (0.50~2.11)		0.939
Diabetes	No * vs. Yes	2.65 (0.89~7.87)		0.080	1.36 (0.48~3.85)		0.560	1.85 (0.55~6.26)		0.323	1.22 (0.37~4.02)		0.747
Hx of breast cancer	No * vs. Yes	1.11 (0.33~3.78)		0.863	1.74 (0.76~3.97)		0.187	2.12 (0.78~5.75)		0.140	1.77 (0.72~4.32)		0.213
FIGO (2018)	IA * vs. IB	1.00 (0.35~2.86)		0.993	1.19 (0.54~2.62)		0.663	1.58 (0.62~4.02)		0.334	1.13 (0.47~2.76)		0.783
	IA * vs. II	1.43 (0.46~4.43)		0.537	1.99 (0.88~4.52)		0.100	1.04 (0.29~3.70)		0.949	2.05 (0.84~4.98)		0.114
Histology	NEEC * vs. G3 EEC	0.21 (0.07~0.63)		0.005	0.89 (0.46~1.71)		0.730	1.17 (0.50~2.69)		0.722	0.61 (0.29~1.29)		0.198
Minimally invasive surgery	No * vs. Yes	0.78 (0.32~0.33)		0.564	1.27 (0.63~2.58)		0.504	0.89 (0.37~2.13)		0.794	1.49 (0.66~3.35)		0.334
Peritoneal washing	No * vs. Yes	2.82 (0.95~8.38)		0.062	1.00 (0.51~1.97)		0.995	1.01 (0.41~2.47)		0.986	1.07 (0.50~2.28)		0.865
Washing result	(−) * vs. (+)	2.79 (0.89~8.77)		0.078	3.50 (1.26~9.76)		0.017	3.16 (0.85~11.80)		0.087	3.58 (1.15~11.13)		0.028
Omentectal evaluation	No * vs. Yes	2.55 (0.92~7.12)		0.073	1.70 (0.77~3.75)		0.189	0.59 (0.14~2.54)		0.476	2.72 (1.19~6.21)		0.018
PaLND	No * vs. Yes	0.70 (0.28~1.74)		0.438	1.20 (0.62~2.30)		0.595	0.60 (0.24~1.48)		0.268	1.08 (0.52~2.24)		0.840
Dissected nodes	<10 * vs. ≥10	0.16 (0.06~0.41)		<0.001	0.70 (0.33~1.49)		0.353	0.57 (0.22~1.49)		0.252	0.47 (0.21~1.04)		0.061
Tumor size		1.05 (0.84~1.32)		0.649	1.02 (0.87 ~1.21)		0.777	0.93 (0.74~1.17)		0.544	0.96 (0.80~1.16)		0.694
DOI/thick		1.06 (1.03~1.09)		<0.001	1.01 (1.00~1.02)		0.065	1.01 (1.00~1.03)		0.130	1.01 (0.99~1.02)		0.293
LVSI	No * vs. Yes	0.81 (0.29~2.27)		0.695	0.87 (0.42~1.81)		0.715	1.68 (0.71~4.00)		0.239	0.85 (0.38~1.93)		0.699
Adjuvant treatment	No * vs. Yes	0.34 (0.14~0.82)		0.016	0.65 (0.34~1.25)		0.197	0.40 (0.17~0.92)		0.031	0.71 (0.34~1.46)		0.346
	No * vs. CTx alone	0.52 (0.15~1.83)		0.306	0.44 (0.13~1.53)		0.198	0.20 (0.03~1.55)		0.124	0.60 (0.17~2.10)		0.420
	No * vs. RTx alone	0.23 (0.07~0.80)		0.021	0.70 (0.33~1.48)		0.351	0.43 (0.16~1.15)		0.091	0.60 (0.25~1.44)		0.253
	No * vs. CTx-RTx	0.43 (0.10~1.94)		0.274	0.72 (0.28~1.84)		0.491	0.48 (0.14~1.71)		0.256	1.06 (0.40~2.81)		0.902
CTx	No * vs. Yes	0.98 (0.38~2.53)		0.962	0.76 (0.37~1.57)		0.457	0.48 (0.16~1.41)		0.181	1.18 (0.55~2.52)		0.677
RTx	No * vs. Yes	0.41 (0.16~1.07)		0.068	0.92 (0.48~1.76)		0.807	0.68 (0.30~1.59)		0.377	0.92 (0.45~1.88)		0.814

* indicates the reference group. Abbreviations: BMI, body mass index; Hx, history; FIGO, International Federation of Gynecology and Obstetrics; NEEC, non-endometroid endometrial cancer; G3 EEC, grade 3 endometrioid carcinoma; PaLND, para-aortic lymph node dissection; DOI, depth of invasion; LVSI, lymphovascular invasion; CTx, chemotherapy; RTx, radiotherapy; OS, overall survival; DFS, disease-free survival; LRRFS, locoregional recurrence-free survival; DMFS, distant metastasis-free survival.

**Table 3 cancers-17-02056-t003:** Multivariate survival analysis.

		OS			DFS			LRRFS			DMFS		
		HR	95% CI	*p* Value	HR	95% CI	*p* Value	HR	95% CI	*p* Value	HR	95% CI	*p* Value
BMI		1.17 (0.95~1.45)		0.147	1.09 (0.96~1.24)		0.177	1.20 (1.03~1.41)		0.021	1.08 (0.95~1.23)		0.250
Age	<60 * vs. ≥60	1.39 (0.31~6.32)		0.667	1.22 (0.46~3.24)		0.691	2.87 (0.76~10.86)		0.120	0.56 (0.17~1.82)		0.331
Diabetes	No * vs. Yes	2.57 (0.62~10.61)		0.192	1.67 (0.47~5.97)		0.429	1.40 (0.31~6.42)		0.663	2.25 (0.52~9.66)		0.277
FIGO (2018)	IA * vs. IB	13.80 (0.65~291.57)		0.092	1.12 (0.20~6.37)		0.901	3.28 (0.24~45.16)		0.375	1.63 (0.23~11.59)		0.625
	IA * vs. II	15.60 (1.09~222.58)		0.043	2.85 (0.60~13.50)		0.188	3.44 (0.25~47.26)		0.356	2.16 (0.36~13.08)		0.402
Histology	NEEC * vs. G3 EEC	0.21 (0.04~1.21)		0.081	0.94 (0.31~2.89)		0.920	2.00 (0.45~8.86)		0.361	0.54 (0.13~2.18)		0.386
Peritoneal washing	No * vs. Yes	4.91 (0.79~30.58)		0.088	0.90 (0.39~2.06)		0.804	1.36 (0.41~4.54)		0.614	0.81 (0.31~2.13)		0.665
Omentectal evaluation	No * vs. Yes	2.08 (0.57~7.59)		0.269	1.86 (0.69~4.99)		0.217	1.00 (0.19~5.42)		1.000	2.06 (0.72~5.95)		0.181
PLND	No * vs. Yes	0.34 (0.08~1.46)		0.146	1.01 (0.27~3.77)		0.995	0.76 (0.13~4.48)		0.759	0.44 (0.11~1.77)		0.245
DOI/thick		0.99 (0.95~1.02)		0.484	1.01 (0.99~1.03)		0.377	1.00 (0.97~1.04)		0.951	1.00 (0.98~1.03)		0.766
Adjuvant treatment	No * vs. Yes	0.41 (0.10~1.61)		0.202	0.65 (0.27~1.56)		0.338	0.40 (0.12~1.29)		0.124	0.70 (0.25~1.92)		0.482
	No * vs. CTx alone	0.34 (0.05~2.32)		0.270	0.58 (0.13~2.57)		0.475	0.47 (0.05~4.82)		0.527	0.85 (0.18~3.96)		0.835
	No * vs. RTx alone	0.72 (0.12~4.47)		0.722	0.71 (0.27~1.84)		0.481	0.32 (0.08~1.29)		0.110	0.66 (0.20~2.17)		0.495
	No * vs. CTx-RTx	0.19 (0.02~1.80)		0.147	0.54 (0.13~2.14)		0.377	0.76 (0.12~4.78)		0.770	0.80 (0.17~3.78)		0.779

* indicates the reference group. Abbreviations: BMI, body mass index; FIGO, International Federation of Gynecology and Obstetrics; NEEC, non-endometroid endometrial cancer; G3 EEC, grade 3 endometrioid carcinoma; PLND, pelvic lymph node dissection; DOI, depth of invasion; CTx, chemotherapy; RTx, radiotherapy; OS, overall survival; DFS, disease-free survival; LRRFS, locoregional recurrence-free survival; DMFS, distant metastasis-free survival.

## Data Availability

The datasets generated during and/or analyzed during the current study are available from the corresponding author on reasonable request.

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
