# Peer review of "High-Risk Early-Stage Endometrial Cancer: Role of Adjuvant Therapy and Prognostic Factors Affecting Survival"

_cancers, 2025, doi:10.3390/cancers17122056_

Round 1

Reviewer 1 Report

Comments and Suggestions for Authors

Dear Authors,

Please find my comments directly on the manuscript file for ease of identification. I look forward to your revised manuscript.

Author Response

Responses to Reviewer 1’s Comments

 1. Summary

Thank you very much for taking the time to review this manuscript, and for your thoughtful and constructive comments. Please find the detailed responses below and the corresponding revisions highlighted changed in the re-submitted files.

2. Point-by-point response to Comments and Suggestions for Authors

Comments 1: Abstract: The FIGO 2018 should be mentioned in the abstract.

Response 1: We appreciate this suggestion. I’ve added FIGO staging in the abstract. (Page 1 line 29)

Comments 2: Methods: Line 96: ‘of hysterectomy to the date of death or disease recurrence’. or last follow-up since not all patients had the event of interest. Also applies to all outcomes.

Response 2: Thank you for pointing this out. We apologize for the omission of this phrase in the manuscript. The relevant sentence has been corrected and added in the Method section. (Page 3 line 102-104)

Overall survival (OS) and disease-free survival (DFS) were evaluated from the date of hysterectomy to the date of death or disease recurrence, respectively, or to the date of last follow-up, if no event occurred.

Comments 3: Line 104-109: ‘All proce-104 dures performed in studies involving human participants were in accordance with the 105 ethical standards of the institutional and/or national research committee and with the Hel-106 sinki Declaration of 1975, as revised in 1983. This study was reviewed and approved by 107 the Institutional Review Board of Seoul St. Mary’s Hospital (Number: KC24RISI0174). Be-108 cause the study was retrospective, the requirement for patient consent was waived.’ This should be at the beginning of the materials and methods.

Response 3: As recommended, the entire ethics statement has been moved to the beginning of the methods section. (Page 2 line 81-86)

The medical records of 345 female patients who underwent hysterectomy for the FIGO stage I and II endometrial cancer at a single institution between September 2008 and December 2022 were reviewed. Patients who were pathologically confirmed to have high-risk non-endometrioid and grade 3 endometrial cancer, were included in the eligibility criteria. Patients diagnosed with grades 1 and 2 endometrial cancer (n= 205), coexisting cancers other than breast and thyroid cancers (n= 11), insufficient records (n=19), and mucinous histology (n=4) were excluded from the analysis. Finally, 106 patients were included in this study. All procedures performed in studies involving human participants were in accordance with the ethical standards of the institutional and/or national research committee and with the Helsinki Declaration of 1975, as revised in 1983. This study was reviewed and approved by the Institutional Review Board of Seoul St. Mary’s Hospital (Number: KC24RISI0174) on March 12, 2024. . Because the study was retrospective, the requirement for patient consent was waived.

Comments 4: Results: Line 114: ‘non-endometrioid endometrial cancer’. The histological subtypes should be presented in detail.

Response 4: Thank you for your comment. The histological subtypes have been added and changed the manuscript. (Page 3 line 118-120)

Among patients with non-endometrioid endometrial cancer, the most common histological type was serous carcinoma, followed by carcinosarcoma, mixed carcinoma, clear cell carcinoma, and undifferentiated carcinoma.

Comments 5: Table 1: It would be easier if you report the numbers together i.e. 46 (43.4%) no separated into 2 columns.

Response 5: Thank you for suggestion. Table 1 has been revised so that each category now lists “n (percentage)” in a single column, as suggested.

Comments 6: Line 142: ‘median follow-up period of 48.8 months’. The calculation method for the follow-up should be added in the statistics chapter of the materials and methods

Response 6: We apologize for missing this detail. I’ve added this in the methods section. Thank you for your comment. (Page 3 line 100-102)

Descriptive statistics were used to analyze the patient characteristics and treatments. These characteristics were compared between non-endometrioid and grade 3 endometrial cancer using Fisher’s exact test and the Mann-Whitney U test. Survival analyses were performed using the Kaplan-Meier method, and the resulting data were evaluated using the log-rank test. The date of origin of all survival outcomes was defined as the hysterectomy date. The follow-up duration for each patient was defined as the time from date of hysterectomy to the date of death, censoring, or last follow-up. The median follow-up was calculated using the reverse Kaplan-Meier method. Overall survival (OS) and disease-free survival (DFS) were evaluated from the date of hysterectomy to the date of death or disease recurrence, respectively, or to the date of last follow-up, if no event occurred. Locoregional recurrence-free survival (LRRFS) was defined as the time from hysterectomy to pelvic or vaginal recurrence. Distant metastasis-free survival (DMFS) was defined as the time from hysterectomy to the occurrence of evidence of distant metastasis.

Comments 7: Figure 2: Please add censored events to the curves. please avoid dotted lines and use full lines of different colors. Colors should be color blind friendly (helpful palettes can be found here coolors.co). For all the curves, including those from the supplementary file.

Response 7: Thank you for suggestion. All figures, including those in the supplementary file, have been revised to display censored events and to use solid lines with color-blind friendly colors ("#0072B2", "#D55E00","#CC79A7"), generated using R programming. Accordingly, the Methods section has also been updated. (Page 3 line 112-113)

Prognostic factors associated with survival outcomes were analyzed using Cox proportional hazards regression models. Multivariate analysis was performed for survival outcomes using backward selection, with covariates that had a p-value of less than 0.1 in the univariate analysis. Statistical significance was set at p < 0.05. All statistical analyses were performed using the STATA/SE software (version 17.0; StataCorp, LLC) and the R program version 4.5.0 (R Development Core Team, Vienna, Austria).

Comments 8: Table 2: Please present HR and 95%CI together instead of in a separated column: 1.09 (0.98-1.22) as an example. Please use 2 decimals for everything instead of the p-value which should be with 3 decimals. When you write no vs yes, it means yes is the reference group, which cannot be given the numbers you report in this analysis. Please correct everywhere and clarify in the captions which is the ref group.

Response 8: Thank you for these suggestions. I’ve changed table 2 and 3 as you suggested, and added footer to clarify the references.

Comments 9: Line 309: ‘molecular characteristics. ‘ You should mention the new FIGO staging which includes the molecular markers as well, which is lacking from the study

Response 9: Thank you for your insightful comments. I fully agree with your suggestion. As noted in the Introduction and Discussion, the wide variation in survival outcomes may be attributed to the histological and molecular heterogeneity of high-grade endometrial cancer, which prompted the FIGO to revise the staging system in 2023. However, molecular profiling such as POLE mutation and MMR status was not routinely performed study period therefore is not included in this analysis. Although we attempted to analyze the data using molecular classification based on TCGA, the available data were insufficient for statistical analysis. As shown in the table below, POLE mutation testing was performed in only 9 patients. Given the importance of these molecular features, as you correctly pointed out, we have now added this limitation to the Discussion section. (Page 11 line 315-318)

Molecular study for NEEC (60)

POLE mutation

N

%

 Performed

9

15.00

 Yes

1

1.67

MMRd

 Performed

25

41.67

 Yes

2

3.33

NSMP

 Performed

46

76.67

 Yes

12

20.00

p53abn

 Performed

46

76.67

 Yes

31

51.67

FIGO Stage 2023

 Stage IAmPOLEmut

1

1.67

 Stage IC

8

13.33

 Stage IIC

20

33.33

 Stage IICmp53abn

31

51.67

This study has several limitations. First, the retrospective nature of the analysis included a heterogeneous cohort of patients, which may affect survival outcomes due to differences in clinical, histological, genetic, and molecular characteristics. Second, the surgical procedures were not standardized. Because of the retrospective nature of the study, patients with preoperative diagnoses suggesting lower-stage with FIGO grade 1-2 or adeno-carcinoma in situ, or with imaging findings indicating disease confined to the uterus without lymph node enlargement, often did not undergo peritoneal washing, omental evaluation, or underwent only limited lymph node dissection. Nevertheless, some of these patients were ultimately diagnosed with high grade early stage endometrial cancer were included in the analysis. Third, heterogeneous treatment modalities can also affect the outcomes, as most treatments are decided by a multidisciplinary team based on individual patient characteristics, preventing randomization of treatment modalities. Fourth, the lack of comprehensive molecular profiling in the real practice clinic limited its analysis based on molecular subtypes. In particular, POLE mutation testing was performed in only a small subset of patients, which restricted evaluation of its prognostic significance. Finally, the statistical power was limited due to the small sample size.

Reviewer 2 Report

Comments and Suggestions for Authors

I read the submitted study with interest. It is an investigation into a very frequent problem in hospital departments: high-grade endometrial cancer, including non-endometrioid and 13 grade 3 endometrioid types. The authors performed a retrospective study on 106 patients, which is a good number of women treated, to draw conclusions and compare with the literature. The results are quite interesting: the major negative prognostic factors are confirmed as 1) lack of lymphadenectomy, 2) omission of adjuvant therapy, 3) deep neoplastic invasion.
However, I need to ask the authors for some specifics, in the M&M:
1) The authors write: Lymph nodal staging, peritoneal washing, and omental evaluations were performed at the discretion of the surgeon.
I would like to know if the surgeon who operated on the women is always the same and with what criteria he chose laparoscopy or robotics (73 women) or open surgery.
2) It is not clear to me whether the surgeon or the surgical staff performed the sentinel lymph node search before performing the lymphadenectomy (for age, weight, comorbidity, etc.?).
3) I am not clear on the criteria according to which the surgical staff chose to perform peritoneal washing and omental evaluation.
4) the number of minimal lymph nodes removed (10) seems too small to me for a sufficient clinical evaluation, since these are cases "at high risk" of recurrence. The studies by Benedetti Panici and Sakuragi identify "at least" 20 lymph nodes as the minimum requirement for pelvic lymphadenectomy.
5) as adjuvant therapy, with what criteria was it decided to perform EBRT or brachytherapy, and the same goes for chemotherapy.
These are fundamental data to better understand the reasoning according to which the study was validly conducted.

Author Response

Comments 1: The authors write: Lymph nodal staging, peritoneal washing, and omental evaluations were performed at the discretion of the surgeon. I would like to know if the surgeon who operated on the women is always the same and with what criteria he chose laparoscopy or robotics (73 women) or open surgery.

Response 1: During the study period (2008–2022), all surgeries were performed by four gynecologic oncology surgeons, each with over 10 years of clinical experience. A multidisciplinary tumor board specializing in gynecologic malignancies was actively operating throughout the period, and surgical principles were consistently shared. There was a clear trend toward minimally invasive surgery over time, i.e. in the early years of the study, open hysterectomy was more common, whereas in later years, laparoscopy and robotic approaches predominated. Additionally, patients with higher FIGO stage (stage II) were more often managed with open surgery compared to those with lower stage disease.

Comments 2: It is not clear to me whether the surgeon or the surgical staff performed the sentinel lymph node search before performing the lymphadenectomy (for age, weight, comorbidity, etc.?).

Response 2: In this patient cohort, which predominantly consisted of early‐stage cases, preoperative imaging rarely demonstrated lymph node enlargement. Consequently, the decision regarding to the extent of lymph node dissection, was made by the operating surgeon based on both preoperative imaging and intraoperative findings. Sentinel lymph node search and lymphadenectomy were both performed by the surgeon at his discretion. In addition to imaging and intraoperative findings, factors such as patient age and performance status were considered as part of the overall risk assessment.

Comments 3: I am not clear on the criteria according to which the surgical staff chose to perform peritoneal washing and omental evaluation.

Response 3: When high‐grade disease was suspected or confirmed preoperatively, peritoneal washing and omental evaluation were routinely performed. However, in most cases where those evaluation were omitted, the preoperative biopsy had indicated a lower grade lesion, such as FIGO grade 1 – 2 or adenocarcinoma in situ, or imaging suggested a clearly confined stage IA tumor. In such instances, high-grade endometrial carcinoma was only identified upon final pathological examination.

Comments 4: the number of minimal lymph nodes removed (10) seems too small to me for a sufficient clinical evaluation, since these are cases "at high risk" of recurrence. The studies by Benedetti Panici and Sakuragi identify "at least" 20 lymph nodes as the minimum requirement for pelvic lymphadenectomy.

Response 4: We appreciate your observation regarding the minimum number of lymph nodes removed. As demonstrated in studies by Benedetti Panici et al. and Sakuragi et al., the removal of at least 20 pelvic nodes has been considered appropriate in high-risk endometrial cancer. In our cohort, patients treated during the earlier years (closer to 2008) generally underwent more extensive lymphadenectomy, frequently with more than 20 nodes. However, over the study period (2008–2022), practice patterns has been slightly changed. For small, apparent stage IA tumors confined to the uterus, surgeons adopted sentinel lymph node (SLN) mapping in accordance with NCCN guidelines. This approach reduced the total number of nodes removed while preserving staging accuracy. Additionally, patients whose pathological diagnosis was underestimated preoperatively often underwent more limited lymph node dissection. Given the extended timeframe of the study, the predominance of early-stage cases, and the nature of retrospective studies, the median nodal count falls below 20. We have now changed a sentence in the Results (Page 3 line 127-129), and added a sentence to the limitation of the Discussion. (Page 11 line 306-312)

Appropriate pelvic lymph node dissection (PLND), defined as the removal of > 10 lymph nodes based on its early stage, was observed at a greater frequency in patients with grade 3 endometrial cancer (60.0% vs. 82.6%, p=0.04) than in those with non-endometrioid endometrial cancer.

This study has several limitations. First, the retrospective nature of the analysis included a heterogeneous cohort of patients, which may affect survival outcomes due to differences in clinical, histological, genetic, and molecular characteristics. Second, the surgical procedures were not standardized. Because of the retrospective nature of the study, patients with preoperative diagnoses suggesting lower-stage with FIGO grade 1-2 or adeno-carcinoma in situ, or with imaging findings indicating disease confined to the uterus without lymph node enlargement, often did not undergo peritoneal washing, omental evaluation, or underwent only limited lymph node dissection. Nevertheless, some of these patients were ultimately diagnosed with high grade early stage endometrial cancer were included in the analysis. Third, heterogeneous treatment modalities can also affect the outcomes, as most treatments are decided by a multidisciplinary team based on individual patient characteristics, preventing randomization of treatment modalities. Fourth, the lack of comprehensive molecular profiling in the real practice clinic limited its analysis based on molecular subtypes. In particular, POLE mutation testing was performed in only a small subset of patients, which restricted evaluation of its prognostic significance. Finally, the statistical power was limited due to the small sample size.

Comments 5: as adjuvant therapy, with what criteria was it decided to perform EBRT or brachytherapy, and the same goes for chemotherapy.

Response 5: When determining the modality of adjuvant radiotherapy, we first consider the extent of lymph node assessment. If a patient has undergone an incomplete pelvic lymph node dissection and the final pathology reveals high‐grade histology, we typically recommend external beam pelvic radiation (EBRT). For stage IA serous carcinoma without myometrial invasion, we prefer intracavitary brachytherapy (ICR) over EBRT. Carcinosarcoma cases are managed with EBRT regardless of myometrial invasion. Molecular markers such as p53 status or other relevant mutations, are also reviewed. Finally, patient‐specific factors including age and performance status, are carefully considered in multidisciplinary discussion before determining whether to add chemotherapy or to escalate or de‐escalate the radiation modality.

EBRT

Brachytherapy

Chemotherapy

FIGO IB, II

No pelvic LN dissection

SND only

FIGO IA serous carcinoma: chemo followed by brachytherapy

Patient factors: multiple comorbidity, old age, RT toxicity, inaccessibility

High risk molecular subtype

Reviewer 3 Report

Comments and Suggestions for Authors

Dear Author,

I reviewed Your article entitled “Early-stage High-grade Endometrial Cancer: Role of Adjuvant therapy and Prognostic Factors Affecting Survival”.

The aim of the article is interesting, as endometrial cancer represents the most common gynecological malignancy and has recently been the focus of extensive research, particularly due to advancements in molecular classification and its prognostic implications. Acquiring a deeper understanding of early-stage high-grade endometrial cancer is essential for guiding optimal treatment decisions and ensuring the best possible patient outcomes.

I have a few suggestions for the Author:

  • Lines 188–190: You state “Additionally, higher body mass index (BMI) (p=0.003), positive washing results (p=0.017), and pathologic omental evaluation (p=0.018) were identified as significant prognostic factors associated with other survival outcomes”. Please clarify which specific survival outcomes are being referred to here.
  • Page numbering: There appears to be an inconsistency in the pagination—page 11 is labeled as page 2 of 15 in the top right corner. Please correct this formatting issue
  • To enhance the scientific value of the work, I suggest adding the following citations.

I suggest citing the following in the introduction section:

  • Restaino S, Paglietti C, Arcieri M, Biasioli A, Della Martina M, Mariuzzi L, Andreetta C, Titone F, Bogani G, Raimondo D, Perelli F, Buda A, Petrillo M, Greco P, Ercoli A, Fanfani F, Scambia G, Driul L, Vizzielli G; The Udine Hospital Gynecological-Oncological Tumor Board Group. Management of Patients Diagnosed with Endometrial Cancer: Comparison of Guidelines. Cancers (Basel). 2023 Feb 8;15(4):1091. doi: 10.3390/cancers15041091. PMID: 36831434; PMCID: PMC9954548.

When speaking about molecular classification I suggest citing the following:

  • Restaino S, Poli A, Arcieri M, Mariuzzi L, Orsaria M, Tulisso A, Pellecchia G, Paparcura F, Petrillo M, Bogani G, Cianci S, Capozzi VA, Biasioli A, Buda A, Mauro J, Fanfani F, Fagotti A, Driul L, Scambia G, Vizzielli G. Molecular classification of endometrial carcinoma on endometrial biopsy: an early prognostic value to guide personalized treatment. Int J Gynecol Cancer. 2024 Aug 5;34(8):1211-1216. doi: 10.1136/ijgc-2024-005478. PMID: 38955372.
  • Occhiali T, Poli A, Arcieri M, Driul L, Ditto A, Bogani G, Mariani A, Scambia G, Restaino S, Vizzielli G. The exciting journey of progress: Exploring FIGO 2023 staging for endometrial cancer at a leading ESGO institution. Eur J Surg Oncol. 2025 Feb 12;51(6):109695. doi: 10.1016/j.ejso.2025.109695. Epub ahead of print. PMID: 40009907.
  • Arcieri M, Vizzielli G, Occhiali T, et al. Application of novel algorithm on a retrospective series to implement the molecular classification for endometrial cancer. Eur J Surg Oncol. 2024 Jul;50(7):108436. doi: 10.1016/j.ejso.2024.108436. Epub 2024 May 23. PMID: 38820923.

When referring to lymph nodal staging cite the following:

  • Cuccu I, Raspagliesi F, Malzoni M, Vizza E, Papadia A, Di Donato V, Giannini A, De Iaco P, Perrone AM, Plotti F, Angioli R, Casarin J, Ghezzi F, Cianci S, Vizzielli G, Restaino S, Petrillo M, Sorbi F, Multinu F, Schivardi G, De Vitis LA, Falcone F, Lalli L, Berretta R, Mueller MD, Tozzi R, Chiantera V, Benedetti Panici P, Fanfani F, Scambia G, Bogani G. Sentinel node mapping in high-intermediate and high-risk endometrial cancer: Analysis of 5-year oncologic outcomes. Eur J Surg Oncol. 2024 Apr;50(4):108018. doi: 10.1016/j.ejso.2024.108018. Epub 2024 Feb 15. PMID: 38428106.
  • Restaino S, Buda A, Puppo A, Capozzi VA, Sozzi G, Casarin J, Gallitelli V, Murgia F, Vizzielli G, Baroni A, Corrado G, Pasciuto T, Ferrari D, Novelli A, Berretta R, Legge F, Vizza E, Chiantera V, Ghezzi F, Landoni F, Scambia G, Fanfani F. Anatomical distribution of sentinel lymph nodes in patients with endometrial cancer: a multicenter study. Int J Gynecol Cancer. 2022 Apr 4;32(4):517-524. doi: 10.1136/ijgc-2021-003253. PMID: 35110375.

The article is well-structured, with the Materials and Methods, Results, and Discussion sections clearly presented. The tables are informative, and the figures effectively represent the text.

It would be interesting to evaluate the molecular classification of the tumors analyzed, as this could potentially influence the survival outcomes. Nonetheless, the results presented are valuable and worthy of publication.

Author Response

Comments 1: Lines 188–190: You state “Additionally, higher body mass index (BMI) (p=0.003), positive washing results (p=0.017), and pathologic omental evaluation (p=0.018) were identified as significant prognostic factors associated with other survival outcomes”. Please clarify which specific survival outcomes are being referred to here.

Response 1: Thank you for your comment. We’ve changed the manuscript as below with clarification of survival outcomes. (Page 6 line 184-185).

Additionally, higher body mass index (BMI) (p=0.003), positive washing results (p=0.017), and pathologic omental evaluation (p=0.018) were identified as significant prognostic factors associated with LRRFS, DFS, and DMFS, respectively.

Comments 2: Page numbering: There appears to be an inconsistency in the pagination—page 11 is labeled as page 2 of 15 in the top right corner. Please correct this formatting issue.

Response 2: Thank you for your comment. Page labelling has been corrected.

Comments 3: I suggest citing the following in the introduction section:

  • Restaino S, Paglietti C, Arcieri M, Biasioli A, Della Martina M, Mariuzzi L, Andreetta C, Titone F, Bogani G, Raimondo D, Perelli F, Buda A, Petrillo M, Greco P, Ercoli A, Fanfani F, Scambia G, Driul L, Vizzielli G; The Udine Hospital Gynecological-Oncological Tumor Board Group. Management of Patients Diagnosed with Endometrial Cancer: Comparison of Guidelines. Cancers (Basel). 2023 Feb 8;15(4):1091. doi: 10.3390/cancers15041091. PMID: 36831434; PMCID: PMC9954548.

Response 3: Added this citation in the introduction section. (Page 2 line 65)

The mainstay of management is surgery, including total hysterectomy and bilateral salpingo-oophorectomy with or without lymph node dissection. Adjuvant therapy is decided based on the pathological evaluation, and radiotherapy (RT) with or without chemotherapy has been recommended for patients with a high risk of recurrence [2,5,6].

Comments 4 When speaking about molecular classification I suggest citing the following:

  • Restaino S, Poli A, Arcieri M, Mariuzzi L, Orsaria M, Tulisso A, Pellecchia G, Paparcura F, Petrillo M, Bogani G, Cianci S, Capozzi VA, Biasioli A, Buda A, Mauro J, Fanfani F, Fagotti A, Driul L, Scambia G, Vizzielli G. Molecular classification of endometrial carcinoma on endometrial biopsy: an early prognostic value to guide personalized treatment. Int J Gynecol Cancer. 2024 Aug 5;34(8):1211-1216. doi: 10.1136/ijgc-2024-005478. PMID: 38955372.
  • Occhiali T, Poli A, Arcieri M, Driul L, Ditto A, Bogani G, Mariani A, Scambia G, Restaino S, Vizzielli G. The exciting journey of progress: Exploring FIGO 2023 staging for endometrial cancer at a leading ESGO institution. Eur J Surg Oncol. 2025 Feb 12;51(6):109695. doi: 10.1016/j.ejso.2025.109695. Epub ahead of print. PMID: 40009907.
  • Arcieri M, Vizzielli G, Occhiali T, et al. Application of novel algorithm on a retrospective series to implement the molecular classification for endometrial cancer. Eur J Surg Oncol. 2024 Jul;50(7):108436. doi: 10.1016/j.ejso.2024.108436. Epub 2024 May 23. PMID: 38820923.

Response 4: Added this citation in the Discussion section. (Page 7 line 237-239)

Although, the importance of molecular profiling has been increasing recognized and re-search in this area is expanding [18-20], its routine implementation in real-world clinical practice remains limited.

Comments 5: When referring to lymph nodal staging cite the following:

  • Cuccu I, Raspagliesi F, Malzoni M, Vizza E, Papadia A, Di Donato V, Giannini A, De Iaco P, Perrone AM, Plotti F, Angioli R, Casarin J, Ghezzi F, Cianci S, Vizzielli G, Restaino S, Petrillo M, Sorbi F, Multinu F, Schivardi G, De Vitis LA, Falcone F, Lalli L, Berretta R, Mueller MD, Tozzi R, Chiantera V, Benedetti Panici P, Fanfani F, Scambia G, Bogani G. Sentinel node mapping in high-intermediate and high-risk endometrial cancer: Analysis of 5-year oncologic outcomes. Eur J Surg Oncol. 2024 Apr;50(4):108018. doi: 10.1016/j.ejso.2024.108018. Epub 2024 Feb 15. PMID: 38428106.
  • Restaino S, Buda A, Puppo A, Capozzi VA, Sozzi G, Casarin J, Gallitelli V, Murgia F, Vizzielli G, Baroni A, Corrado G, Pasciuto T, Ferrari D, Novelli A, Berretta R, Legge F, Vizza E, Chiantera V, Ghezzi F, Landoni F, Scambia G, Fanfani F. Anatomical distribution of sentinel lymph nodes in patients with endometrial cancer: a multicenter study. Int J Gynecol Cancer. 2022 Apr 4;32(4):517-524. doi: 10.1136/ijgc-2021-003253. PMID: 35110375.

Response 5: Added this citation in the Discussion section. (Page 10 line 278)

In this study, FIGO stage II was an unfavorable prognostic factor for OS, consistent with a population-based cohort study by Akesson et al. [28] Significant controversy persists regarding the extent of lymph node dissection and the role of sentinel lymph node (SLN) biopsy in early-stage endometrial cancer. In our study, lymphadenectomy involving > 10 lymph nodes and the depth of invasion were significant predictive factors for OS in the univariate analysis. Some studies supports SLN mapping followed by full lymphadenectomy [29-32], and the NCCN guidelines recommend SLN biopsy for the surgical staging of endometrial cancer regardless of the risk group. However, concerns regarding its safety remain because of its therapeutic role. The retrospective SEPAL study showed a survival benefit for systemic pelvic and para-aortic lymphadenectomy in the high-risk group [33], and the ESMO-ESGO-ESTRO guidelines support lymphadenectomy as part of comprehensive staging [6].

Comments 6: The article is well-structured, with the Materials and Methods, Results, and Discussion sections clearly presented. The tables are informative, and the figures effectively represent the text.

It would be interesting to evaluate the molecular classification of the tumors analyzed, as this could potentially influence the survival outcomes. Nonetheless, the results presented are valuable and worthy of publication.

Response 6: Thank you for your insightful comments. I fully agree with your suggestion. As noted in the Introduction and Discussion, the wide variation in survival outcomes may be attributed to the histological and molecular heterogeneity of high-grade endometrial cancer, which prompted the FIGO to revise the staging system in 2023. However, molecular profiling such as POLE mutation and MMR status was not routinely performed study period therefore is not included in this analysis. Although we attempted to analyze the data using molecular classification based on TCGA, the available data were insufficient for statistical analysis. As shown in the table below, POLE mutation testing was performed in only 9 patients. Given the importance of these molecular features, as you correctly pointed out, we have now added this limitation to the Discussion section. (Page 11 line 315-318)

Molecular study for NEEC (60)

POLE mutation

N

%

 Performed

9

15.00

 Yes

1

1.67

MMRd

 Performed

25

41.67

 Yes

2

3.33

NSMP

 Performed

46

76.67

 Yes

12

20.00

p53abn

 Performed

46

76.67

 Yes

31

51.67

FIGO Stage 2023

 Stage IAmPOLEmut

1

1.67

 Stage IC

8

13.33

 Stage IIC

20

33.33

 Stage IICmp53abn

31

51.67

This study has several limitations. First, the retrospective nature of the analysis included a heterogeneous cohort of patients, which may affect survival outcomes due to differences in clinical, histological, genetic, and molecular characteristics. Second, the surgical procedures were not standardized. Because of the retrospective nature of the study, patients with preoperative diagnoses suggesting lower-stage with FIGO grade 1-2 or adeno-carcinoma in situ, or with imaging findings indicating disease confined to the uterus without lymph node enlargement, often did not undergo peritoneal washing, omental evaluation, or underwent only limited lymph node dissection. Nevertheless, some of these patients were ultimately diagnosed with high grade early stage endometrial cancer were included in the analysis. Third, heterogeneous treatment modalities can also affect the outcomes, as most treatments are decided by a multidisciplinary team based on individual patient characteristics, preventing randomization of treatment modalities. Fourth, the lack of comprehensive molecular profiling in the real practice clinic limited its analysis based on molecular subtypes. In particular, POLE mutation testing was performed in only a small subset of patients, which restricted evaluation of its prognostic significance. Finally, the statistical power was limited due to the small sample size.

Reviewer 4 Report

Comments and Suggestions for Authors

Dear Authors,

I was very pleased to review this interesting manuscript about early stage high-grade endometrial cancer and the potential factors that affect survival.

The methodology and the statistical methods that were used in the manuscript are of high quality. However, there are some concerns:

1) There are no data about the molecular classification, only in table 1 the percentage of p53abn. However, the information only for this bio marker and not for POLEmut and MMRstatus adds no value in the manuscript, since the complete molecular profile should be known. So, I think this information should be removed or data about POLEmut and MMRstatus should be added.

2) To establish that the patients were early stage (I-II), adequate lymph node staging should be performed. Especially for high-risk patients (as in this study) SLNB or pelvic and para-aortic lymphadenectomy should be performed. However, in adequate lymph node staging has been performed only in 41.5%. So, some patients might not be truly early stage and some stage III patients might be missed. Therefore, survival data might be affected by the worst survival of advanced stages (that might be not treated with the correct adjuvant treatment). Patient that did not receive correct lymph node staging should be removed.

3) In the results and conclusions, lack of lymphadenectomy should not be presented as a reason for worse survival, because these patients did not receive the correct treatment and it’s not ethically correct to present these data.

Author Response

Comments 1: There are no data about the molecular classification, only in table 1 the percentage of p53abn. However, the information only for this biomarker and not for POLEmut and MMRstatus adds no value in the manuscript, since the complete molecular profile should be known. So, I think this information should be removed or data about POLEmut and MMRstatus should be added.

Response 1: Thank you for your insightful comments.

As noted in the Introduction and Discussion, the wide variation in survival outcomes may be attributed to the histological and molecular heterogeneity of high-grade endometrial cancer. However, molecular profiling such as POLE mutation and MMR status was not routinely performed study period therefore is not included in this analysis. Although we attempted to analyze the data using molecular classification based on TCGA, the available data were insufficient for statistical analysis. As shown in the table below, POLE mutation testing was performed in only 9 patients. Given the importance of these molecular features, we have now added this limitation to the Discussion section. (Page 11 line 315-318)

Molecular study for NEEC (60)

POLE mutation

N

%

 Performed

9

15.00

 Yes

1

1.67

MMRd

 Performed

25

41.67

 Yes

2

3.33

NSMP

 Performed

46

76.67

 Yes

12

20.00

p53abn

 Performed

46

76.67

 Yes

31

51.67

FIGO Stage 2023

 Stage IAmPOLEmut

1

1.67

 Stage IC

8

13.33

 Stage IIC

20

33.33

 Stage IICmp53abn

31

51.67

This study has several limitations. First, the retrospective nature of the analysis included a heterogeneous cohort of patients, which may affect survival outcomes due to differences in clinical, histological, genetic, and molecular characteristics. Second, the surgical procedures were not standardized. Because of the retrospective nature of the study, patients with preoperative diagnoses suggesting lower-stage with FIGO grade 1-2 or adeno-carcinoma in situ, or with imaging findings indicating disease confined to the uterus without lymph node enlargement, often did not undergo peritoneal washing, omental evaluation, or underwent only limited lymph node dissection. Nevertheless, some of these patients were ultimately diagnosed with high grade early stage endometrial cancer were included in the analysis. Third, heterogeneous treatment modalities can also affect the outcomes, as most treatments are decided by a multidisciplinary team based on individual patient characteristics, preventing randomization of treatment modalities. Fourth, the lack of comprehensive molecular profiling in the real practice clinic limited its analysis based on molecular subtypes. In particular, POLE mutation testing was performed in only a small subset of patients, which restricted evaluation of its prognostic significance. Finally, the statistical power was limited due to the small sample size.

In addition, although our current study did not demonstrate a statistically significant difference in outcomes related to p53abn status, we acknowledge that previous studies have identified p53abn as an important prognostic marker affecting survival outcomes in endometrial cancer [1-3]. Accordingly, p53abn status was taken into consideration when determining adjuvant therapy in clinical practice. Based on this context, the inclusion of information on p53abn may enhance the reader’s understanding of the manuscript.

[1] Whelan K, et al. TP53 mutation and abnormal p53 expression in endometrial cancer: Associations with race and outcomes. Gynecol Oncol. 2023 Nov;178:44-53. doi: 10.1016/j.ygyno.2023.09.009. Epub 2023 Sep 23. PMID: 37748270.

[2] Jia H, et al.. p53 Immunohistochemistry staining patterns and prognosis significance in 212 cases of non-endometrioid endometrial cancer. Pathol Res Pract. 2024 Nov;263:155595. doi: 10.1016/j.prp.2024.155595. Epub 2024 Sep 18. PMID: 39316989.

[3] Talhouk A, et al. Confirmation of ProMisE: A simple, genomics-based clinical classifier for endometrial cancer. Cancer. 2017 Mar 1;123(5):802-813. doi: 10.1002/cncr.30496. Epub 2017 Jan 6. PMID: 28061006.,

Comments 2: To establish that the patients were early stage (I-II), adequate lymph node staging should be performed. Especially for high-risk patients (as in this study) SLNB or pelvic and para-aortic lymphadenectomy should be performed. However, in adequate lymph node staging has been performed only in 41.5%. So, some patients might not be truly early stage and some stage III patients might be missed. Therefore, survival data might be affected by the worst survival of advanced stages (that might be not treated with the correct adjuvant treatment). Patient that did not receive correct lymph node staging should be removed.

Response 2: Thank you for this important observation. We fully agree that adequate lymph node staging is necessary to accurately determine early-stage disease, particularly in high-risk endometrial cancer patients. As the reviewer correctly notes, inadequate lymph node assessment may lead to understaging, and consequently, affect survival outcomes due to the inclusion of occult stage III cases.

In our study, adequate pelvic lymph node dissection (defined as removal of >10 nodes) was performed in 41.5% of patients. While this number falls short of the threshold suggested in studies such as those by Benedetti Panici and Sakuragi, this reflects real-world clinical practice over a long study period (2008–2022). Notably, surgical practice evolved over time, sentinel lymph node (SLN) mapping was increasingly adopted for patients with apparent stage IA disease in accordance with NCCN guidelines. In addition, preoperative biopsy in many cases suggested low-grade histology (e.g., FIGO grade 1–2 or adenocarcinoma in situ), or imaging showed disease confined to the uterus without lymph node enlargement, resulting in more conservative surgical approaches. However, some of these patients were ultimately diagnosed with high-grade endometrial cancer on final pathology and were thus included in our analysis.

We acknowledge that the variability in lymph node assessment may have introduced bias. For transparency, we have added a statement in the Results to clarify the proportion of patients undergoing adequate lymph node dissection and a corresponding note in the Discussion under limitations (Page 11, line 306-312). However, we respectfully believe that excluding these patients entirely would not accurately reflect real-world clinical decision-making, and that our findings still provide meaningful insight into outcomes for high-grade endometrial cancer across clinical practice.

This study has several limitations. First, the retrospective nature of the analysis included a heterogeneous cohort of patients, which may affect survival outcomes due to differences in clinical, histological, genetic, and molecular characteristics. Second, the surgical procedures were not standardized. Because of the retrospective nature of the study, patients with preoperative diagnoses suggesting lower-stage with FIGO grade 1-2 or adeno-carcinoma in situ, or with imaging findings indicating disease confined to the uterus without lymph node enlargement, often did not undergo peritoneal washing, omental evaluation, or underwent only limited lymph node dissection. Nevertheless, some of these patients were ultimately diagnosed with high grade early stage endometrial cancer were included in the analysis. Third, heterogeneous treatment modalities can also affect the outcomes, as most treatments are decided by a multidisciplinary team based on individual patient characteristics, preventing randomization of treatment modalities. Fourth, the lack of comprehensive molecular profiling in the real practice clinic limited its analysis based on molecular subtypes. In particular, POLE mutation testing was performed in only a small subset of patients, which restricted evaluation of its prognostic significance. Finally, the statistical power was limited due to the small sample size.

Comments 3: In the results and conclusions, lack of lymphadenectomy should not be presented as a reason for worse survival, because these patients did not receive the correct treatment and it’s not ethically correct to present these data.

Response 3: Thank you again for your thoughtful comment. We agree that patients who did not receive adequate lymph node staging may have been undertreated, and this may have impacted their clinical outcomes. However, we respectfully disagree with the suggestion that presenting these data is ethically inappropriate. First, our study reflects real-world clinical practice over fourteen years, during which surgical standards, staging methods, and adjuvant treatment strategies evolved. As such, variability in lymphadenectomy was not due to intentional omission of care, but rather based on preoperative clinical assessments, surgeon discretion, and institutional practice at the time. Second, we present the absence of lymphadenectomy as a prognostic factor, acknowledging that patients who did not undergo full staging may have been understaged or undertreated, potentially contributing to poorer outcomes. This is stated in the Discussion and Limitations sections of the manuscript. Third, omitting such patients from the analysis would introduce selection bias and diminish the generalizability of our findings. Including these patients allows us to present the challenges and consequences of staging variability, particularly in high-grade endometrial cancer. However, as we understand your thoughtful consideration, we have ensured to describe this in the limitation of revised manuscript (Page 11, line 306-318).

Round 2

Reviewer 1 Report

Comments and Suggestions for Authors

Thank you for going through my comments. I am afraid I missed an important point in my previous revision: The classification in NEEC and grade 3 does not make sense to me as some of the NEEC you describe can also have grade 3. so please clearly present the patients in low grade vs high grade and add the histotypes in detail in table 1.

Also adjust the names throughout the paper and figures as appropiate.

Author Response

Comment:

Thank you for going through my comments. I am afraid I missed an important point in my previous revision: The classification in NEEC and grade 3 does not make sense to me as some of the NEEC you describe can also have grade 3. so please clearly present the patients in low grade vs high grade and add the histotypes in detail in table 1.

Also adjust the names throughout the paper and figures as appropiate.

Responses to Reviewer 1’s Comments:

Thank you for your comments.

The classification used in this manuscript was intended to compare aggressive histological types of non-endometrioid endometrial carcinoma (NEEC) such as serous, clear cell, undifferentiated, mixed, and carcinosarcoma with high-grade (grade 3) endometrioid carcinoma (EEC)( as described in the abstract, and on page 1). Therefore, as you suggested, we’ve clarified all tables, figures, and throughout the paper including supplement tables (highlighted). Also the histologic types have been added in Table 1.

Reviewer 4 Report

Comments and Suggestions for Authors

Response 1 comments:

Thank you for your response. Since the time period that was studied molecular classification was not widely used, I recommend that the authors remove the data about p53 status, because there are no available data about the POLE and MMR status, so multiple classifiers might be missed, leading to biased results.

Response 2 comments:

Thank you for your response. The reviewer respectfully disagrees that 40% of lymph node staging is presenting real world data, since the majority of the patients did not receive at all lymphadenectomy (not just a lower number of lymph nodes dissected. At what percentage low grade histology at biopsy was upgraded to high grade histology in the hysterectomy specimen?

Response 3 comments:

“variability in lymphadenectomy was not due to intentional omission of care, but rather based on preoperative clinical assessments, surgeon discretion, and institutional practice at the time”: Please explain me the logic behind omission of lymphadenectomy based on scientific data (guidelines) that the surgeons and the institutions where following at that time of period that proposed no lymph node staging for high-risk endometrial cancer.

New comment: The title should be changed to “high-risk instead of high-grade endometrial cancer.

Author Response

Comments 1: 

Thank you for your response. Since the time period that was studied molecular classification was not widely used, I recommend that the authors remove the data about p53 status, because there are no available data about the POLE and MMR status, so multiple classifiers might be missed, leading to biased results.

Response 1:

Thank you again for your recommendation. We removed p53 status in Table 1 and Table 2.

Comments 2:

Thank you for your response. The reviewer respectfully disagrees that 40% of lymph node staging is presenting real world data, since the majority of the patients did not receive at all lymphadenectomy (not just a lower number of lymph nodes dissected. At what percentage low grade histology at biopsy was upgraded to high grade histology in the hysterectomy specimen?

Response 2:

Thank you for your consideration. We’ve attached the table for details on the result for biopsy and imaging evaluation. Details on this patients are described Table below. 

NEEC (n=60)

G3 Endometrioid (n=46)

No

PLND

15 (25.0%)

-> 11 pts showed underestimated bx results

3         (6.5%)

-> 2 pts showed underestimated bx results

#

bx

Image and details

Bx

result

1

Endometrioid FIGO grade 1

MRI evaluation FIGO IA

1: Adenocarcinoma

1: Endometrioid FIGO Grade 2

1: Endometrioid FIGO grade 3

5

Endometrioid FIGO grade 2

MRI evaluation FIGO IA

1

Endometrioid FIGO grade 3

 done SLN

1

Adenocarcinoma in situ

 done SLN

3

Adenocarcinoma

MRI evaluation FIGO IA

 done SLN

2

clear cell carcinoma

(age 82) – image finding: subtle enhancing lesion (<5mm), cN0

 done SLN

2

carcinosarcoma

(age 87) – image finding: No enlarged lymph nodes

 done SLN only

Comment 3: 

“variability in lymphadenectomy was not due to intentional omission of care, but rather based on preoperative clinical assessments, surgeon discretion, and institutional practice at the time”: Please explain me the logic behind omission of lymphadenectomy based on scientific data (guidelines) that the surgeons and the institutions where following at that time of period that proposed no lymph node staging for high-risk endometrial cancer.

Response 3: 

Thank you for your consideration. From the ESMO guideline 2022,

(1) Sentinel LNE can be considered as a strategy for nodal assessment in low-risk/intermediate-risk EC (e.g. stage IA G1-G3 and stage IB G1-G2) [II, A]. It can be omitted in cases without myometrial invasion. Systematic LNE is not recommended in this group [II, D].

(2) Surgical lymph node staging should be carried out in patients with high-intermediate-risk/high-risk disease. Sentinel lymph node biopsy is an acceptable alternative to systematic LNE for lymph node staging in high-intermediate-risk/high-risk stage I-II [III, B].

Also in the ESMO guideline 2010,

“Hysterectomy with bilateral salpingo-oophorectomy ± bilateral pelvic-para-aortic lymphadenectomy” for stage IA and IB grade 1-3 endometrioid endometrial cancer."

In this study, 87% of patients were stage I, and among them, 72% was stage IA.

Therefore, patients whose disease underestimated before surgery (for instance, endometrioid carcinoma FIGO grade 1 with MRI evaluation of FIGO IA) underwent sentinel lymph node biopsy without PLND or, as shown in table 1, had fewer than 10 lymph nodes removed during operation based on surgeons’ discretion.

Comment 4: 

New comment: The title should be changed to “high-risk instead of high-grade endometrial cancer.

Response 4:

Thank you for the reviewer’s thoughtful comment. Since this study exclusively includes high-grade histologic subtypes—grade 3 endometrioid carcinoma, serous, clear cell, carcinosarcoma, and undifferentiated carcinoma— we initially thought that the title, “Early-stage High-grade Endometrial Cancer,” accurately reflected the study population.

We understand that the suggestion to revise the title to “High-Risk Early-Stage Endometrial Carcinoma” instead of “High-grade” stems from the fact that molecular classification, including p53 status, was not applied in this study. Therefore, we have revised the title to “High-Risk Early-Stage Endometrial Carcinoma.” (highlighted).

Round 3

Reviewer 1 Report

Comments and Suggestions for Authors

Thank you for the modifications

For figure 1: the label "Grade 3 EEC" is missaligned; please fix this issue. Also please change the label in the risk table as well.

Best,

Author Response

Comments:

For figure 1: the label "Grade 3 EEC" is missaligned; please fix this issue. Also please change the label in the risk table as well.

Response:

Thank you for your comments. I’ve modified misalignment, and the risk table in figure 1.

Reviewer 4 Report

Comments and Suggestions for Authors

The authors for the first time mention that a SLNB has been performed to the patients. This is a vital information and should be added at the results and in Table 1.

Author Response

Comment:

The authors for the first time mention that a SLNB has been performed to the patients. This is a vital information and should be added at the results and in Table 1.

Response:

Thank you for your comments. I’ve added the SLNB in Table 1 and results with highlight. (page 3 line 127-129)

Sentinel lymph node biopsy was performed in 17 (28.3%) with non-endometrioid endometrial cancer and in 11 (23.9%) patients with grade 3 endometrioid carcinoma.

Table 1 Patient and Tumor characteristics

Characteristic

Total (n=106)

NEEC (n=60)

G3 EEC (n=46)

P-value

SLNB

28 (26.4%)

17 (28.3%)

11 (23.9%)

0.662